# Training in the Initial Range of Motion Promotes Greater Muscle Adaptations Than at Final in the Arm Curl

**DOI:** 10.3390/sports11020039

**Published:** 2023-02-06

**Authors:** Gustavo F. Pedrosa, Marina G. Simões, Marina O. C. Figueiredo, Lucas T. Lacerda, Brad J. Schoenfeld, Fernando V. Lima, Mauro H. Chagas, Rodrigo C. R. Diniz

**Affiliations:** 1Weight Training Laboratory, School of Physical Education, Physiotherapy and Occupational Therapy, Federal University of Minas Gerais, Lagoa Santa 31270-901, Brazil; 2Brazilian Air Force, Aeronautical Instruction and Adaptation Center, Lagoa Santa 33400-000, Brazil; 3Department of Physical Education, State University of Minas Gerais, Divinópolis 35501-170, Brazil; 4Department of Health Sciences, CUNY Lehman College, Bronx, New York, NY 10468, USA

**Keywords:** muscle hypertrophy, partial angular, partial range of motion, partial angular displacement, muscle strength

## Abstract

Objective: The effects of ROM manipulation on muscle strength and hypertrophy response remain understudied in long-term interventions. Thus, we compared the changes in strength and regional muscle hypertrophy after training in protocols with different ranges of motion (ROM) in the seated dumbbell preacher curl exercise using a within-participant experimental design. Design and methods: Nineteen young women had one arm randomly assigned to train in the initial ROM (INITIAL_ROM_: 0°–68°; 0° = extended elbow) while the contralateral arm trained in the final ROM (FINAL_ROM_: 68°–135°), three times per week over an eight-week study period. Pre- and post-training assessments included one repetition maximum (1RM) testing in the full ROM (0°–135°), and measurement of biceps brachii cross-sectional area (CSA) at 50% and 70% of humerus length. Paired *t*-tests were used to compare regional CSA changes between groups, the sum of CSA changes at 50% and 70% (CSA_summed_), and the strength response between the training protocols. Results: The INITIAL_ROM_ protocol displayed a greater CSA increase than FINAL_ROM_ protocol at 70% of biceps length (*p* = 0.001). Alternatively, we observed similar increases between the protocols for CSA at 50% (*p* = 0.311) and for CSA_summed_ (*p* = 0.111). Moreover, the INITIAL_ROM_ protocol displayed a greater 1RM increase than FINAL_ROM_ (*p* < 0.001). Conclusions: We conclude that training in the initial angles of elbow flexion exercise promotes greater distal hypertrophy of the biceps brachii muscle in untrained young women. Moreover, the INITIAL_ROM_ condition promotes a greater dynamic strength increase when tested at a full ROM compared to the FINAL_ROM_.

## 1. Introduction

The effects of manipulating resistance training (RT) variables on muscle hypertrophy response has been an ongoing focus of investigation [1,2,3]. Among the RT variables, range of motion (ROM) was generally overlooked in past studies targeting training prescription recommendations [3,4]. However, its influence on neuromuscular responses is becoming increasingly recognized as a potential area of research interest [5,6]. ROM may alter the length at which the working muscle contracts. At the beginning of a concentric action (INITIAL_ROM_), the muscle length is longer than at the final angles of the action (FINAL_ROM_). Several investigations have reported greater metabolic stress and IGF-1 release after contractions performed at longer vs. shorter muscle lengths [7,8,9,10,11], which have been associated with muscle hypertrophy [12,13]. This raises the possibility that training exclusively in the INITIAL_ROM_, where the muscle is in a lengthened state, may promote a greater hypertrophic response than training in the FINAL_ROM_.

Several studies have compared the hypertrophic response to training in the INITIAL_ROM_ and FINAL_ROM_ [7,14,15]. For the quadriceps, results showed greater increases in muscle cross-sectional area (CSA) at the distal muscle regions of the rectus femoris [14] and vastus lateralis [7,14] for the group that performed the INITIAL_ROM_ compared to the FINAL_ROM_ training group. These findings suggest that the ROM trained may influence regional muscle growth on quadriceps muscle. Similar to results for the quadriceps, Sato et al. [16] found a greater increase in muscle thickness of the distal biceps brachii plus brachialis after preacher unilateral arm curl resistance training at INITIAL_ROM_ compared to FINAL_ROM_. However, the training occurred two times per week across five weeks, raising the question as to whether differences would be maintained after a longer training period. According to Halperin et al. [17], it is possible that the initial improvements were due to the novel stimulus rather than an inherent superiority of the program. Conceivably, results may have been different if the study period lasted longer. Therefore, it is necessary to conduct further research to provide greater insight into the effect of training in various ROMs on the muscle hypertrophic response over longer time frames.

Kassiano et al. [15] compared changes in muscle thickness of the medial and lateral heads of the gastrocnemius across an eight-week RT program involving the calf raise exercise when performed in the INITIAL_ROM_ vs FINAL_ROM_. Results showed the INITIAL_ROM_ training elicited greater increases in muscle thickness of the medial and lateral gastrocnemius compared to the FINAL_ROM_, providing evidence that training exclusively at longer muscle lengths may be superior to training exclusively at shorter muscle lengths for enhancing muscle hypertrophy of the plantar flexors. However, these authors assessed hypertrophy at a single point along the muscle, and it is known that resistance training in different ROM configuration may induce non-homogeneous hypertrophic adaptations across the length of the muscle [14,18]. Furthermore, the change in muscle volume analyzed via magnetic resonance imaging (MRI) is considered the gold standard for estimating muscle hypertrophy [19]. However, this procedure is expensive [19]. A more economical alternative to analyzing muscle volume by MRI is to assess muscle CSA by B-mode ultrasound [19], with measurements taken at different sites along the length of the muscle [14,20]. This approach provides a proxy for muscle volume, imparting a better comprehension of the influence of ROM manipulation on muscle hypertrophy.

In addition to its effects on muscle hypertrophy, RT also promotes increases in maximal strength. Research generally shows that strength increases are specific to the given ROM trained [21,22]. Accordingly, training in the INITIAL_ROM_ and FINAL_ROM_ could lead to a greater strength increase in the specific angles trained [23]. Taking into account that the sticking point is a primary determinant of performance in the 1RM test [24], it can be expected that training in the INITIAL_ROM_, which promotes greater strength increases in this region, would also cause greater increases in the 1RM test performed in full ROM in comparison to training in the FINAL_ROM_. In line with this reasoning, Pedrosa et al. [14] found that 12 weeks training in the INITIAL_ROM_ promoted a greater 1RM increase at full ROM than training in the FINAL_ROM_ using the knee extension machine. Werkhausen et al. [2] compared the isokinetic strength after training in the INITIAL_ROM_ and the full ROM in the leg press exercise. Results showed similar changes between training in the INITIAL_ROM_ and a full ROM, suggesting the strength adaptations across a full ROM are influenced by the initial training angle. These findings further the rationale that training in the INITIAL_ROM_ could be superior to training in the FINAL_ROM_ in a full ROM strength test. However, this hypothesis has not yet been objectively tested in other muscles such as the elbow flexors.

Therefore, this study aimed to compare dynamic strength and regional muscle hypertrophy changes of the elbow flexors in young women after performing the arm curl exercise for eight weeks while training in INITIAL_ROM_ and FINAL_ROM_. We hypothesized that the INITIAL_ROM_ protocol would elicit greater increases in dynamic strength and distal muscle hypertrophy than FINAL_ROM_ protocol.

## 2. Methods

We employed a within-participant experimental design whereby the right and left arms of 21 untrained women trained in INITIAL_ROM_ or FINAL_ROM_ for eight weeks. We chose the seated dumbbell preacher curl exercise to ensure that participants maintained strict form throughout exercise performance. Pre- and post-study strength was assessed by the 1RM test in the arm curl, and B-mode ultrasound was employed to assess CSA changes in the mid- and distal regions of the biceps.

The sample size was estimated a priori following the recommendations of Beck [25] using the software G*Power (version 3.1.9.2; Heinrich Heine Universität Düsseldorf, DE, Germany). We used the t-statistic with an alpha of 0.05, power of 0.8, and a relatively moderate Cohen’s d effect size (ES) of 0.7 and determined that 19 subjects were required for adequate statistical power. We recruited two additional subjects to account for the possibility of dropouts, expected to be ~10% of the sample.

Participants were untrained women who had not performed any physical activity for at least six months prior to the onset of the study. Two participants withdrew for personal reasons; therefore, 19 women completed the study (mean age = 22.8 ± 10.5 years; mean body mass = 64.5 ± 8.05 kg; mean height = 164.1 ± 4.7 cm). Each participant’s upper limb was allocated in a randomized fashion according to upper limb dominance. The order of training was counterbalanced whereby half of the participants performed the INITIAL_ROM_ protocol with their preferred limb, while the other half performed the FINAL_ROM_ protocol with their preferred limb. Before participation, written consent was obtained from each participant after being informed of the procedures, risks, and benefits of the investigation. The study followed the standards established in the Declaration of Helsinki and was approved by the ethics committee of the Federal University of Minas Gerais (approval #CAAE 91438418.4.0000.5149).

In the first pre-training session, after the anthropometric data assessment, we obtained measures of biceps brachii CSA at 50% and 70% of the distance from the acromion to the lateral epicondyle of each humerus via B-mode ultrasound (MindRay^®^ DC-7, Shenzhen, China). It should be noted that muscle hypertrophy assessed by B-mode ultrasound imaging is highly correlated with MRI, which is considered the gold standard for measuring changes in muscle mass [26]. Images were acquired at a frequency of 21 frames/s, using a 4–10 MHz linear transducer with a depth ranging from 1 to 6 cm and gain between 50 and 64 db. The settings were individually adjusted to produce a clear image of the entire muscle for extended field-of-view, and replication at post-training. The same trained technician performed all ultrasound scans, moving the transducer in a line parallel to the humeral epicondyles at a relatively constant speed for approximately 7s at each site. The images were saved to hard drive and coded for blinded CSA calculation using the Horos^®^ software. We averaged the two CSA measurements in each region to obtain the final values used for analyses. Moreover, we summed the CSA at 50%, and 70% of biceps length (CSA_summed_) to produce an estimate of overall biceps hypertrophy. Previous studies have employed similar formulas in an attempt to produce a hypertrophy measure more representative of the whole muscle in comparison to a single muscle region [1,20]. The intraclass correlation coefficients (ICC_3,1_) in our laboratory for CSA at 50% and 70% of biceps length were 0.94 and 0.92, respectively.

After ultrasound imaging, we assessed participants 1RM in the seated dumbbell preacher curl exercise. The 1RM test was performed alternately on each arm throughout a full ROM, with a 3-min recovery interval between the limbs and between attempts. A final value was obtained within 5 attempts on each arm. Each attempt started with the elbow fully extended (0°), and the shoulder angle (humerus and trunk) fixed at 45°. The dumbbell was handed to the participant in this initial position, who then performed a concentric muscle action until 135° of elbow flexion (forearm perpendicular to the ground). The attempt was considered successful if the participant was able to perform the full range of elbow flexion (0° to 135°) without assistance from auxiliary movements. The dumbbell load was progressively increased (minimum of 0.5 kg) until the participant was unable to perform the concentric action with proper form. Hence, the 1RM value corresponded to the weight lifted in the previous successful attempt. This initial test was considered a familiarization session to the 1RM assessment. The 1RM test was repeated 48 h later (pre-training session 2) and the value obtained in this session was used for statistical analysis. Between 48 h and 72 h after the last training session, the ultrasound imaging and the 1RM test in a full ROM were repeated using the procedures previously described.

Training sessions consisted of the seated dumbbell preacher curl performed in a specific ROM for each limb. The shoulder angle was fixed at 45° (as in the 1RM test). The INITIAL_ROM_ protocol was trained from 0° to 68° of elbow flexion, and the FINAL_ROM_ protocol was trained from 68° to 135° of elbow flexion (Figure 1). The ROMs were individually checked by a manual goniometer (axis fixed in the elbow, and the rules fixed in the arm and forearm) at the beginning of each training session. An elastic cord was placed in front of the machine to serve as a mechanical stop ensuring training was limited to the prescribed ROMs (Figure 1). In the INITIAL_ROM_, participants began the eccentric muscle action when their forearm touched the string (68° of elbow flexion) and they continued lowering the load until full extension. In the FINAL_ROM_ protocol, the participants began the concentric action when their forearm touched the string (68° of elbow flexion) and they continued raising the load until their forearm was perpendicular to the ground (135° of elbow flexion).

Each protocol was trained three times per week, in the same session, separated by 48–72 h over the eight-week study period. We alternated the session-to-session order in which the training protocols were performed: i.e., if the INITIAL_ROM_ protocol was the first to be performed in a given training session, the FINAL_ROM_ protocol would be trained first in the next session. If the participant was assigned to start with INITIAL_ROM_ protocol, after completing each set, 1 min interval was allowed before initiating the FINAL_ROM_ protocol with the contra lateral limb. The next set for the starting protocol was only initiated after 3 min of the completion of the previous set of the INITIAL_ROM_; all recovery periods were timed with a stopwatch to ensure accuracy.

Participants performed four sets per session in both the INITIAL_ROM_ and the FINAL_ROM_ protocols. In an effort to standardize the training stimulus for the development of hypertrophy and muscle strength, all sets were carried out until volitional failure [27,28]. When the last set was performed with more than 10 or less than 8 repetitions, the load was increased or reduced diminished in 1 kg at the next training session, respectively. Each repetition was performed with a 2 s concentric action and a 2 s eccentric action (timed by metronome). Five sets were performed from the fifth week on, following the same previously described procedures.

The Shapiro–Wilk test confirmed the normality of data distribution, and all variables presented similar baseline values between training protocols. We analyzed the absolute difference values (post–pre-values) between training protocols by paired *t*-test for all variables of interest. We reported 95% confidence intervals (CI) around the point estimate. Cohen’s d effect sizes (ES) were calculated (post-pre/pooled standard deviation) with the following interpretation: trivial: <0.20; small: 0.20–0.60; moderate: 0.61–1.20; large: 1.21–2.0; very large: >2.0) [29]. All statistical procedures were performed using JASP statistics packages, version 0.14 (Wagenmakers, Amsterdam). We considered statistical significance when α < 0.05.

## 3. Results

When comparing regional CSA between the training protocols, analysis showed the INITIAL_ROM_ protocol displayed a greater CSA increase than the FINAL_ROM_ protocol at 70% of biceps brachii length (*p* = 0.001; 95% CI = 0.18 to 0.59 cm^2^; ES = 0.89), and a relatively similar CSA increase at 50% (*p* = 0.331; 95% CI = −0.10 to 0.34 cm^2^; ES = 0.23). Analysis showed the CSA_summed_ increase was not statistically different between the training protocols (*p* = 0.111; 95% CI = −0.08 to 0.67 cm^2^; ES = 0.39), as shown in Figure 2. Regarding the 1RM test, analysis showed the INITIAL_ROM_ protocol presented a statistically greater increase than the FINAL_ROM_ protocol (*p* < 0.001; 95% CI = 0.39 to 1.06 kg; ES = 1.05), as shown in Figure 3.

## 4. Discussion

A primary finding of our study was that training in the INITIAL_ROM_ elicited greater increases in CSA at 70% of biceps brachii length and in the 1RM test than the FINAL_ROM_ protocol. These results are consistent with our initial hypothesis and provide further evidence that ROM manipulation impacts regional muscular adaptations across a variety of different muscles and exercises.

To our knowledge, only three previous studies compared regional hypertrophic changes after training in INITIAL_ROM_ and FINAL_ROM_. McMahon et al. [7] found that the INITIAL_ROM_ group achieved greater vastus lateralis hypertrophy than the FINAL_ROM_ group only at the distal region after 8 weeks of knee extension training. Similarly, Pedrosa et al. [14] showed the INITIAL_ROM_ group presented greater distal muscle growth of the rectus femoris and the vastus lateralis muscles than the FINAL_ROM_ group after 12 weeks of knee extension training. Moreover, Sato et al. [16] demonstrated the INITIAL_ROM_ training elicited greater distal biceps brachii plus brachialis muscle hypertrophy after only five weeks resistance training. These results corroborate our findings, which suggest that training in INITIAL_ROM_ promotes greater distal hypertrophy of the biceps than training in FINAL_ROM_, and a similar hypertrophy response between conditions at the middle region in young, untrained women. Moreover, our study expands on the findings of McMahon et al. [7] and Pedrosa et al. [14] and supports those of Sato et al. [16] by providing evidence that training a muscle at long muscle length has a beneficial effect on muscular adaptations in the upper extremities.

Some other studies have investigated regional muscle hypertrophy between the FINAL_ROM_ and the FULL_ROM_ protocols [14,18,31]. Bloomquist et al. [18] reported greater hypertrophy in the middle and distal regions of the anterior quadriceps femoris in a group of young men performing the back squat for 12 weeks in a FULL_ROM_ group compared to the FINAL_ROM_ group. Similarly, McMahon et al. [31] reported greater distal vastus lateralis hypertrophy in a group of young men and women performing a variety of lower limb exercises in a FULL_ROM_ versus a FINAL_ROM_ over an 8-week training period. Pedrosa et al. [14] showed greater distal muscle hypertrophy of rectus femoris and vastus lateralis muscles after training in FULL_ROM_ compared to FINAL_ROM_. Given that the main difference between training in FINAL_ROM_ and FULL_ROM_ is that the FULL_ROM_ excurses the INITIAL_ROM_, it can be speculated that the greater muscle hypertrophy after training in a FULL_ROM_ results from training at longer muscle lengths.

Although speculative, a possible mechanistic explanation for the heightened regional hypertrophic response is related to the production of higher amounts of metabolic stress [10], and insulin-like growth factor (IGF)-1 release [7] when training at longer muscle lengths in comparison to training at shorter muscle lengths, which in turn may confer anabolic effects [32]. Additionally, there is evidence suggesting that both metabolic stress [33] and IGF-1 [34] concentrations may vary between muscle regions after mechanical overload, and that greater regional muscle hypertrophy occurs in regions demonstrating greater metabolic stress [35] and IGF-1 concentrations [34]. Thus, we hypothesize that training in the INITIAL_ROM_ promotes a heightened physiological response at the distal portion of the muscle, thereby leading to greater muscle protein increase in this region. Previous research supports our findings [7,14,18,31]; however, no attempts were made to explore mechanisms involved, which requires further investigation.

When summing the CSA results of the two regional sites (CSA_summed_), hypertrophic increases were statistically similar between training protocols. This value provides a general proxy for hypertrophy of the muscle as a whole. Our findings in this regard are consistent with previous studies on the topic [1,20]. However, although the study lasted eight weeks and is in line with previous research that aimed to measure muscle hypertrophy over this time period [7,31], we cannot necessarily infer that results would hold true over longer-term interventions nor rule out the possibility that other factors may have influenced changes [36]. Therefore, further investigation on topic using longer interventional period is recommended to confirm or refute the present findings.

In regard to the 1RM results, our study shows that training in the INITIAL_ROM_ elicits greater dynamic strength improvements in a full ROM test compared to training in the FINAL_ROM_; these results were observed despite the use of heavier absolute loads when training in the FINAL_ROM_. To our knowledge, only one study to date has compared changes in dynamic strength at a FULL_ROM_ after training in the INITIAL_ROM_ versus FINAL_ROM_. Pedrosa et al. [14] reported the INITIAL_ROM_ group showed greater 1RM test increase at a FULL_ROM_ compared to the FINAL_ROM_ after 12 weeks of knee extension training, lending support to our results.

Several studies indicate a ROM-specific strength increase after training in a FULL_ROM,_ FINAL_ROM_ [18,21,22], and INITIAL_ROM_ [14]. Bloomquist et al. [18] found that the training groups (FULL_ROM_ and FINAL_ROM_) presented greater 1RM increases in the trained ROM. Similarly, Martínez-Cava et al. [21] reported ROM-specific strength adaptations after 10 weeks of training in the bench press exercise at a full ROM, two-thirds ROM, and one-third-ROM). In addition, Pedrosa et al. [14] showed the INITIAL_ROM_ and FINAL_ROM_ groups presented greater 1RM increases in the ROM trained. Although the present study did not compare the strength performance in different ROMs, previous findings [14,18,21,22,23] support the rationale that a ROM-specific strength increase may also have occurred in the present study, and therefore influenced the results of the 1RM test in a full ROM. Accordingly, the INITIAL_ROM_ training would allow a greater strength enhancement at the beginning of the concentric action of a full ROM compared to training in the FINAL_ROM_ [14]. We speculate that this specific joint-angle strength adaptation (from the INITIAL_ROM_ protocol) was fundamental to overcoming the sticking point and thus resulted in a superior increase in 1RM.

Changes in muscle morphology may help to provide a mechanistic explanation for the observed angular specific differences in strength increase between conditions [37]. Evidence indicates a positive association between greater increases in distal muscle CSA regions and increases in torque angles where the muscle is elongated [37]. Thus, the greater muscle hypertrophy response at 70% in the INITIAL_ROM_ condition may have enhanced strength performance in a full ROM 1RM test to a greater extent than training in FINAL_ROM_. This hypothesis needs further investigation.

Additionally, it is known that the increase in maximum dynamic strength is related to alterations in neural factors, such as an increase in the number of activated motor units, improvement in inter and intra muscular synchronization, and a reduction in the activation of the antagonist muscles during exercise performance [38]. Thepaut-Mathieu et al. [39] reported that resistance training performed exclusively at a short muscle length promoted greater neural adaptations (as assessed by surface electromyography) near or at the trained angles, and this adaptation coincided with the greater observed increases in muscle strength near or at the trained angles; the response did not occur in the group that trained in long muscle length. This finding is supported by Noorkoiv et al. [37], who demonstrated a positive and significant correlation between the force increase at short muscle length angles and the increase of the surface electromyographic signal only in the group that trained at the short muscle length compared to the long length group. It therefore could be hypothesized that the contribution from the improvement of neural factors to increase the performance of the 1RM test was smaller in the FINAL_ROM_ group compared to INITIAL_ROM_ group. The present study did not attempt to assess neural factors; hence, further investigations are needed to better elucidate the mechanisms of strength increases associated with ROM manipulation.

This study has several limitations that should be taken into account when attempting to draw practical conclusions. First, although performing sets to volitional fatigue helps to ensure that all individuals receive a comparable hypertrophic stimulus [27], its implementation influences other variables, such as the number of repetitions performed. Future studies should seek to study ROM-induced muscular adaptations with different configurations of RT variables. Second, we only tested dynamic strength in a full ROM using the knee extension machine and thus cannot extrapolate findings to specific partial ROMs, isometric strength at different joint-angles or the transfer of these results to the activities of everyday life. Third, the findings are specific to dynamic elbow flexion exercise and thus cannot necessarily be generalized to other exercises or muscle groups. Fourth, we measured hypertrophy at only two points along the length of the biceps brachii; it would be interesting to analyze other regions or even the muscle volume to obtain a more robust perspective of hypertrophic changes. Fifth, our findings are specific to young, healthy, untrained women and thus cannot necessarily be generalized to other populations.

Finally, we chose to employ a within-participant design, which affords the benefit of enhancing statistical power by reducing the amount of between-participant variability [40]. While this experimental model can provide keen insights into skeletal muscle adaptations in longitudinal RT investigations [40], a potential limitation of this design is the possibility of a cross-education effect. There is evidence indicating that the cross-education effect, if it indeed occurs, would be restricted to neural parameters and muscle strength gains; morphological changes (e.g., CSA) are not materially influenced by this effect [41]. Hence, any muscle strength gains achieved in the contralateral limb should conceivably evolve from an increase in motor neuron activation without contribution from morphological adaptations. Moreover, previous studies investigating the cross-education effect for EMG amplitude have shown inconclusive results [42,43]. For example, Hortobágyi, et al. [42] found that changes in the EMG amplitude of the untrained limb depend on the training mode performed (e.g., type of muscle action). The neuromuscular changes were similar to the changes in muscle strength. Moreover, other researchers found that the cross-education effect contributes to approximately 7.8% of the muscle strength gain of the contralateral limb [44]. In the present study, the mean relative increase in the INITIAL_ROM_ and FINAL_ROM_ groups was 42.8 ± 14.8% and 19.0 ± 10.3%, respectively; these differences would be outside of any potential confounding from cross-education. Furthermore, it has been argued that, when both limbs of an individual are trained with different protocols, the cross-education effect is minimal or non-existent [44,45]. Thus, based on the magnitude of influence presented by Munn et al. [44], it seems likely that any difference in the strength responses between limbs would be due to training protocols with minimal confounding from cross-education.

## 5. Conclusions

In conclusion, the seated dumbbell preacher curl performed in the INITIAL_ROM_ elicited greater biceps brachii hypertrophy at 70% length, but not at 50% nor for the CSA_summed_ in untrained women. Moreover, the INITIAL_ROM_ protocol promoted a greater increase in the 1RM test in full ROM compared with training in the FINAL_ROM_. These findings add to the body of literature indicating that training at long muscle lengths promotes increases in hypertrophy at the distal muscle region, and these findings occur in both the upper and lower extremity musculature.

## Figures and Tables

**Figure 1 sports-11-00039-f001:**
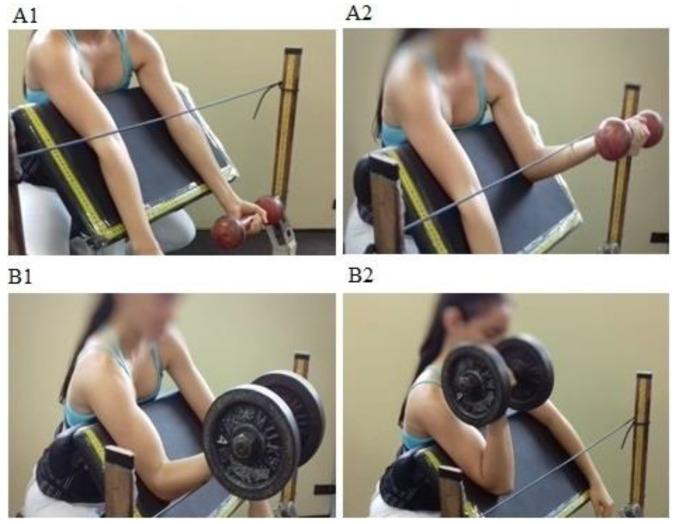
Training protocols range of motion. (**A1**,**A2**) = starting and finishing the concentric action of INITIAL_ROM_ protocol (0°–68° of elbow flexion), respectively. (**B1**,**B2**) = starting and finishing the concentric action of FINAL_ROM_ protocol (68°–135° of elbow flexion), respectively.

**Figure 2 sports-11-00039-f002:**
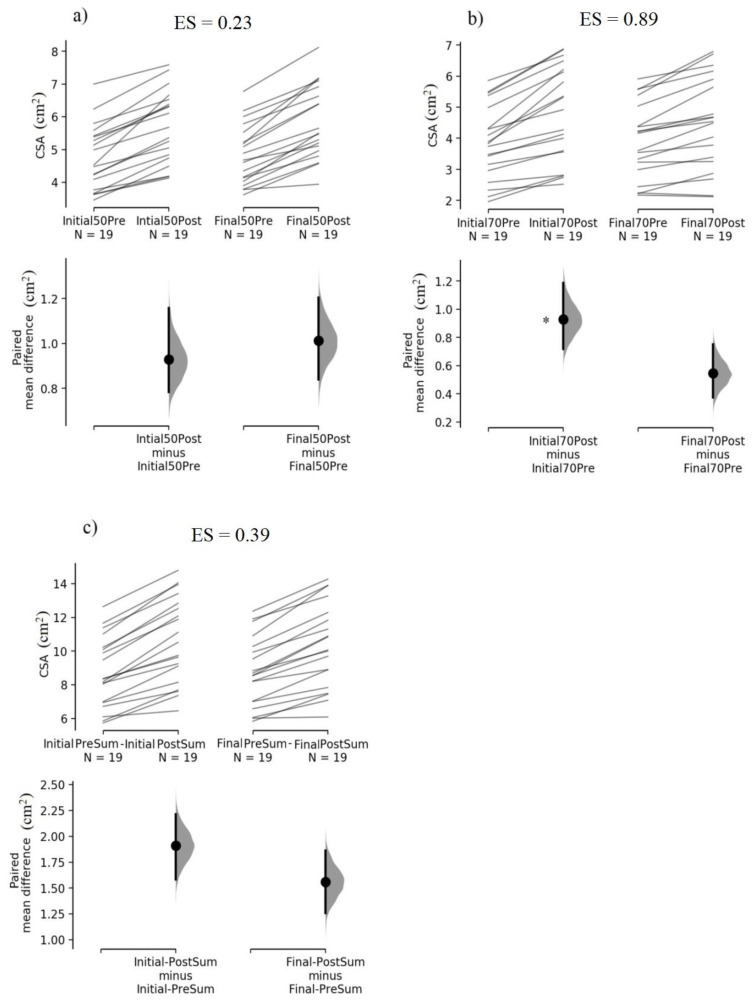
The paired mean difference for cross-sectional area in INITIAL_ROM_ and FINAL_ROM_ at (**a**) 50% humeral length; (**b**) 70% humeral length; and (**c**) summed values of 50% and 70% humeral length [30]. The raw data are plotted on the upper axes; each paired set of observations is connected by a line. On the lower axes, each paired mean difference is plotted as a bootstrap sampling distri-bution. Mean differences are depicted as dots; 95% confidence intervals are indicated by the ends of the vertical error bars. ES = effect size. * Significant differences compared with FINALROM protocol.

**Figure 3 sports-11-00039-f003:**
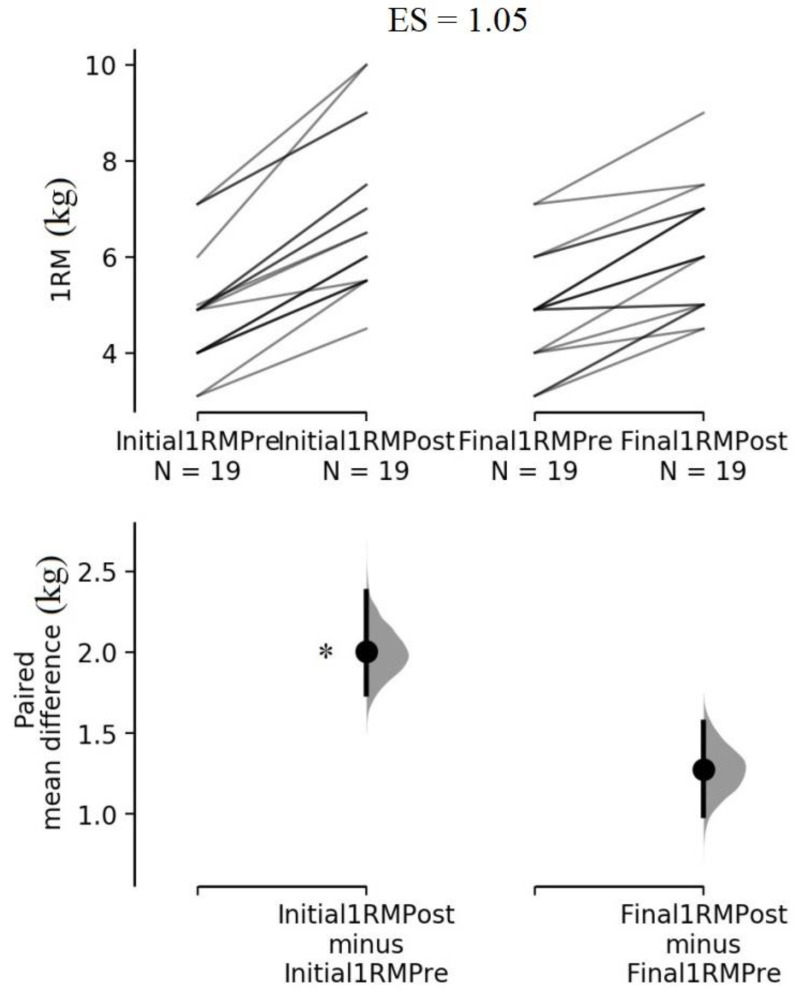
The paired mean difference for 1 repetition maximum in INITIAL_ROM_ and FINAL_ROM_ [30]. The raw data are plotted on the upper axes; each paired set of observations is connected by a line. On the lower axes, each paired mean difference is plotted as a bootstrap sampling distribution. Mean differences are depicted as dots; 95% confidence intervals are indicated by the ends of the vertical error bars. ES = effect size. * Significant differences compared with FINAL_ROM_ protocol.

## Data Availability

All data generated or analyzed during this study will are included in the published article as Table(s) and Figure(s). Any other data requirement can be directed to the corresponding author upon reasonable request.

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
