# Peer review of "Training in the Initial Range of Motion Promotes Greater Muscle Adaptations Than at Final in the Arm Curl"

_sports, 2023, doi:10.3390/sports11020039_

Round 1

Reviewer 1 Report

General Comments:

I have read with interest and critical thinking the proposed manuscript “Training in the initial range of motion promotes greater muscle adaptations than at final in the arm curl”. I believe it is useful do add applied studies to unravel what practically works in a real scenario when it comes to hypertrophy and strength.

Notes for the specific comments:

Some comments where I believe it is appropriate to suggest changes are highlighted using this formatting: “example”. This can mean: addition, removal, rewriting.

The document I have been provided with, did not have line numbers associated, hence, I have been unable to provide easy locations of the text. I tried to refer as line 1 the first line of each page of the main text only (including section heading where applicable).

Specific Comments:

Abstract:

Page 1, line 3: Design and Methods. D should be bold “D

Keywords: partial range of motion is a repetition in the title. I suggest partial amplitude, partial angular displacement, etc.

Introduction:

Page 1, line 21-22: It reads a bit strange this sentence:  One variable that has not received much attention in research is the range of motion (ROM) [3,4]. Because you have referenced several studies where there is indeed research in the range of motion. Either remove it or be more precise because referencing contradicts the statement. There is evidence from both fundamental and applied research about the effect of training at different ROM on adaptations.

Methods:

Page 4, line 11: could you explicitly justify why volitional failure was required and provide evidence for the need of it? “until volitional failure”

Results:

Please add the effect sizes in all figures as we expect figures to be read independently and correctly interpreted based on all relevant statistical parameters. 

Discussion:

Page 7, line 31 onwards. I appreciate the reflections, but it seems appropriate to include neural mechanisms, they did not receive any attention. You justify 1RM performance with peripheral adaptations only, but maximal strength notoriously is also highly dependent on neural features.

In, the next paragraph, discussing limitations, I’m wondering whether the use of contralateral limb, as a different condition, also impacts on 1RM performance (i.e. confounding factor due to the ability of the nervous system to adapt even if one side of the body would not be used. In this case, from that perspective it would be like to receive double stimulus?). Essentially, for the 1RM performance, is the research design a limitation? This should be discussed aligned with the relevant body of literature.

Author Response

Reviewer 1:

We thank the reviewer for your detailed comments. We have provided point-by-point responses to your comments below, and where applicable have revised the manuscript with text highlighted in yellow.

  • Page 1, line 3: Design and Methods. D should be bold “D

AUTHOR RESPONSE: Revised as you recommended.

  • Keywords: partial range of motion is a repetition in the title. I suggest partial amplitude, partial angular displacement, etc.

AUTHOR RESPONSE: Revised as you recommended (page 2 – line 23-24)

3) Page 1, line 21-22: It reads a bit strange this sentence:  One variable that has not received much attention in research is the range of motion (ROM) [3,4]. Because you have referenced several studies where there is indeed research in the range of motion. Either remove it or be more precise because referencing contradicts the statement. There is evidence from both fundamental and applied research about the effect of training at different ROM on adaptations.

AUTHOR RESPONSE: We revised the sentence to improve clarity. Please see the changes in Pag 3 Line 3-5

“Among the training variables, range of motion (ROM) was overlooked in past studies targeting training prescription recommendations [3,4]. But its influence on neuromuscular responses is currently recognized and has been focus of debated [5,6].”

4) Page 4, line 11: could you explicitly justify why volitional failure was required and provide evidence for the need of it? “until volitional failure”

AUTHOR RESPONSE: It is known that the induction of muscle hypertrophy or strength increase is not dependent on training being performed until muscle failure (Grgic et al., 2021). However, it is also unclear in the literature how protocols with different levels of proximity to failure will affect regional muscle hypertrophy responses. Previous studies have shown that training at different volumes and intensities when performed until muscle failure tends to provide a similar hypertrophy (Jenkins et al., 2016; Nobrega et al., 2018). Thus, we chose it in the present study in an attempt to equate the training stimulus via muscle failure, even considering the limitation that the protocols were performed at different volumes and intensities. Therefore, in line with these previous findings and the recommendations of Dankel et al. (2017) for the use of failure as a means for standardization, we tried to minimize the heterogeneity between training protocols and their influence on neuromuscular responses leading training to failure.

To your point, we have revised and inserted references to justify why training occur until failure. Please see page 9 line 1-2.

“All sets were carried out until volitional failure in an effort to standardize the training stimulus for the development of hypertrophy and muscle strength [27,28].”

Dankel, S. J., Jessee, M. B., Mattocks, K. T., Mouser, J. G., Counts, B. R., Buckner, S. L., & Loenneke, J. P. (2017). Training to Fatigue: The Answer for Standardization When Assessing Muscle Hypertrophy?. Sports medicine (Auckland, N.Z.)47(6), 1021–1027. https://doi.org/10.1007/s40279-016-0633-7

Grgic, J., Schoenfeld, B. J., Orazem, J., & Sabol, F. (2022). Effects of resistance training performed to repetition failure or non-failure on muscular strength and hypertrophy: A systematic review and meta-analysis. Journal of sport and health science11(2), 202–211. https://doi.org/10.1016/j.jshs.2021.01.007

Jenkins, N. D., Housh, T. J., Buckner, S. L., Bergstrom, H. C., Cochrane, K. C., Hill, E. C., Smith, C. M., Schmidt, R. J., Johnson, G. O., & Cramer, J. T. (2016). Neuromuscular Adaptations After 2 and 4 Weeks of 80% Versus 30% 1 Repetition Maximum Resistance Training to Failure. Journal of strength and conditioning research30(8), 2174–2185. https://doi.org/10.1519/JSC.0000000000001308

Nóbrega, S. R., Ugrinowitsch, C., Pintanel, L., Barcelos, C., & Libardi, C. A. (2018). Effect of Resistance Training to Muscle Failure vs. Volitional Interruption at High- and Low-Intensities on Muscle Mass and Strength. Journal of strength and conditioning research, 32(1), 162–169. https://doi.org/10.1519/JSC.0000000000001787

5) Please add the effect sizes in all figures as we expect figures to be read independently and correctly interpreted based on all relevant statistical parameters. 

AUTHOR RESPONSE: We added the ES to the figures as requested. Please see at pag 20 and 21

6) Page 7, line 31 onwards. I appreciate the reflections, but it seems appropriate to include neural mechanisms, they did not receive any attention. You justify 1RM performance with peripheral adaptations only, but maximal strength notoriously is also highly dependent on neural features.

- We revised to address your suggestion. Please see at page 13 and line 7-25

“Additionally, it is known that the increase in maximum dynamic strength is related to alterations in neural factors, such as an increase in the number of activated motor units, improvement in inter and intra muscular synchronization, and a reduction in the activation of the antagonist muscles during exercise performance [37]. Thepaut-Mathieu et al. [38], reported that resistance training performed exclusively at a short muscle length promoted greater neural adaptations (as assessed by surface electromyography) near or at the trained angles, and this adaptation coincided with the greater observed increases in muscle strength near or at the trained angles; the response did not occur in the group that trained in long muscle length. This finding is supported by Noorkoiv et al. [36], who demonstrated a positive and significant correlation between the force increase at short muscle length angles and the increase of the surface electromyographic signal only in the group that trained at the short muscle length compared to the long length group. It therefore could be hypothesized that the contribution from the improvement of neural factors to increase the performance of the 1RM test was smaller in the FINALROM group compared to INITIALROM group. The present study did not attempt to assess neural factors; hence, further investigations are needed to better elucidate the mechanisms of strength increases associated with ROM manipulation..”

7) In, the next paragraph, discussing limitations, I’m wondering whether the use of contralateral limb, as a different condition, also impacts on 1RM performance (i.e. confounding factor due to the ability of the nervous system to adapt even if one side of the body would not be used. In this case, from that perspective it would be like to receive double stimulus?). Essentially, for the 1RM performance, is the research design a limitation? This should be discussed aligned with the relevant body of literature.

AUTHOR RESPONSE: You raise an interesting point as to the within-participant design. Given that the unilateral exercise model reduces inter-subject variability, it can serve to substantially enhance statistical power and thus provide greater ability to draw strong causal inferences (MacInnis et al., 2017). Finally, and most important to our study, it has been argued that, when both limbs of an individual are trained with different protocols, the cross-education effect is minimal or non-existent (Bell et al., 2020; Munn et al., 2004). Hence, we surmise that any difference in the strength responses between limbs would be due to training protocols and not influenced by a cross-education effect (Fisher et al., 2016).

MacInnis MJ, McGlory C, Gibala MJ, Phillips SM. 2017. Investigating human skeletal muscle physiology with unilateral exercise models: when one limb is more powerful than two. Applied Physiology, Nutrition, and Metabolism = Physiologie Appliquee, = Nutrition et Metabolisme 42(6):563_570 DOI 10.1139/apnm-2016-0645.

Bell ZW, Wong V, Spitz RW, Chatakondi RN, Viana R, Abe T, Loenneke JP. 2020. The contraction history of the muscle and strength change: lessons learned from unilateral training models. Physiological Measurement 41(1):01TR01 DOI 10.1088/1361-6579/ab516.

Munn J, Herbert RD, Gandevia SC. 2004. Contralateral effects of unilateral resistance training: a meta-analysis. Journal of Applied Physiology 96(5):1861_1866 DOI 10.1152/japplphysiol.00541.2003.

Fisher J, Blossom D, Steele J. 2016. A comparison of volume-equated knee extensions to failure, or not to failure, upon rating of perceived exertion and strength adaptations. 41(2):168_174 DOI 10.1139/apnm-2015-0421.

To address your point, we have amended the limitations as follows.

“Finally, we chose to employ a within-participant design, which affords the benefit of enhancing statistical power by reducing the amount of between-participant variability [39]. While this experimental model can provide keen insights into skeletal muscle adaptations in longitudinal RT investigations [39],  a potential limitation of this design is the possibility of a cross-education effect. There is evidence indicating that the cross-education effect, if it indeed occurs, would be restricted to neural parameters and muscle strength gains; morphological changes (e.g., CSA) are not materially influenced by this effect [40]. Hence, any muscle strength gains achieved in the contralateral limb should conceivably evolve from an increase in motor neuron activation without contribution from morphological adaptations. Moreover, previous studies investigating the cross-education effect for EMG amplitude have shown inconclusive results [41,42]. For example, Hortobágyi, et al. [41] found that changes in the EMG amplitude of the untrained limb depend on the training mode performed (e.g., type of muscle action). The neuromuscular changes were similar to the changes in muscle strength. Moreover, other researchers found that the cross-education effect contributes to approximately 7.8% of the muscle strength gain of the contralateral limb [43]. In the present study, the mean relative increase in the INITIALROM and FINALROM groups were 42.8±14.8% and 19.0±10.3%, respectively; these differences would be outside of any potential confounding from cross-education. Furthermore, it has been argued that, when both limbs of an individual are trained with different protocols, the cross-education effect is minimal or non-existent [43,44]. Thus, based on the magnitude of influence presented by Munn et al. [43], it seems likely that any difference in the strength responses between limbs would be due to training protocols with minimal confounding from cross-education.”.

Bell ZW, Wong V, Spitz RW, Chatakondi RN, Viana R, Abe T, Loenneke JP. 2020. The contraction history of the muscle and strength change: lessons learned from unilateral training models. Physiological Measurement 41(1):01TR01 DOI 10.1088/1361-6579/ab516.

Beyer KS, Fukuda DH, Boone CH, Wells AJ, Townsend JR, Jajtner AR, Gonzalez AM, Fragala MS, Hoffman JR, Stout JR. 2016. Short-term unilateral resistance training results in cross education of strength without changes in muscle size, activation, or endocrine response. Journal of Strength and Conditioning Research 30(5):1213_1223 DOI 10.1519/JSC.0000000000001219.

Fisher J, Blossom D, Steele J. 2016. A comparison of volume-equated knee extensions to failure, or not to failure, upon rating of perceived exertion and strength adaptations. 41(2):168_174 DOI 10.1139/apnm-2015-0421.

Hortobágyi T, Lambert NJ, Hill JP. 1997. Greater cross education following training with muscle lengthening than shortening. Medicine and Science in Sports and Exercise 29(1):107_112.

Lee M, Carroll TJ. 2007. Cross education: possible mechanisms for the contralateral effects of unilateral resistance training. Sports Medicine 37(1):1_14 DOI 10.2165/00007256-200737010-00001.

Munn J, Herbert RD, Gandevia SC. 2004. Contralateral effects of unilateral resistance training: a meta-analysis. Journal of Applied Physiology 96(5):1861_1866 DOI 10.1152/japplphysiol.00541.2003.

Reviewer 2 Report

Dear Authors,

Thank you for the opportunity to read and review the manuscript entitled "Training in the initial range of motion promotes greater muscle adaptations than at final in the arm curl". Overall, the topic is suitable for sports and interesting for the readers. Interesting aspects can also be derived for sports practice. 

Some suggestions for improvement: 

In the abstact the subheadings should not be written bold. Furthermore, the font and font size should be unified (see also the introduction). Furthermore, it should be emphasized even more that effects are poorly addressed in long-term interventions (central research question). 

Although the studies were conducted for 8 weeks, longer-term hypertrophy effects cannot be explained. Therefore, this fact should be discussed more. 

B-mode ultrasound can be used to determine muscle thickness. However, the gold standard would be Xray. This should also be discussed more in detail. 

The text formatting on page 4 should be changed. 

The limitations should be expanded to include the points of muscle cross-section determination and purely dynamic force testing. Since the muscle is usually moved via functional ROM during everyday activities, a limitation of the findings for everyday life should be addressed. 

Author Response

Reviewer 2:

We thank the reviewer for your detailed comments. We have provided point-by-point responses to your comments below, and where applicable have revised the manuscript with text highlighted in yellow.

1) In the abstact the subheadings should not be written bold. Furthermore, the font and font size should be unified (see also the introduction). Furthermore, it should be emphasized even more that effects are poorly addressed in long-term interventions (central research question). 

AUTHOR RESPONSE:  Thanks for pointing out these issues. As per your suggestion, we removed the bold and adjusted the font, as well emphasizing the central research question. Please see it at page 2

“Objective: The effects of ROM manipulation on muscle strength and hypertrophy response remain poorly elucidated in long-term interventions.”

2) Although the studies were conducted for 8 weeks, longer-term hypertrophy effects cannot be explained. Therefore, this fact should be discussed more. 

AUTHOR RESPONSE: Good point. Accordingly, revised to address your suggestion. Please see it at page 11 – starting in line 22

“However, although the study lasted eight weeks and is in line with previous research that aimed to measure muscle hypertrophy over this time period [7,30], we cannot necessarily infer that results would hold true over longer-term interventions nor rule out the possibility that other factors may have influenced changes [35]. Therefore, further investigation on topic using longer interventional period is recommended to confirm or refute the present findings.”

3) B-mode ultrasound can be used to determine muscle thickness. However, the gold standard would be Xray. This should also be discussed more in detail. 

AUTHOR RESPONSE: We assume you meant magnetic resonance imaging as the gold standard to measure muscle hypertrophy (Stokes et al., 2021). While true, we would note that B-mode ultrasound is recognized as a valid tool to measure muscle size change (Stokes et al., 2021; Scott et al., 2017) and in fact displays a high correlation with MRI for the measurement of hypertrophy. We added this information in the text. Please, see page 6 line 18-19

“It should be noted that muscle hypertrophy assessed by B-mode ultrasound imaging is highly correlated with MRI [25].”

Stokes, T., Tripp, T. R., Murphy, K., Morton, R. W., Oikawa, S. Y., Lam Choi, H., McGrath, J., McGlory, C., MacDonald, M. J., & Phillips, S. M. (2021). Methodological considerations for and validation of the ultrasonographic determination of human skeletal muscle hypertrophy and atrophy. Physiological reports9(1), e14683. https://doi.org/10.14814/phy2.14683

Scott, J. M., Martin, D. S., Ploutz-Snyder, R., Matz, T., Caine, T., Downs, M., Hackney, K., Buxton, R., Ryder, J. W., & Ploutz-Snyder, L. (2017). Panoramic ultrasound: a novel and valid tool for monitoring change in muscle mass. Journal of cachexia, sarcopenia and muscle8(3), 475–481. https://doi.org/10.1002/jcsm.12172

4) The text formatting on page 4 should be changed. 

AUTHOR RESPONSE: We revised as requested.

5) The limitations should be expanded to include the points of muscle cross-section determination and purely dynamic force testing. Since the muscle is usually moved via functional ROM during everyday activities, a limitation of the findings for everyday life should be addressed.    

AUTHOR RESPONSE: Fair point. We revised to address your suggestion. Please, see page 14-15

“We only tested dynamic strength in a full ROM using the knee extension machine and thus cannot extrapolate findings to specific partial ROMs, isometric strength in different joint-angles or the transfer of these results to the activities of everyday life.”
